# The Role of Autophagy and Apoptosis in Neuropathic Pain Formation

**DOI:** 10.3390/ijms23052685

**Published:** 2022-02-28

**Authors:** Ming-Feng Liao, Kwok-Tung Lu, Jung-Lung Hsu, Chih-Hong Lee, Mei-Yun Cheng, Long-Sun Ro

**Affiliations:** 1Linkou Medical Center, Department of Neurology, Chang Gung Memorial Hospital, College of Medicine, Chang Gung University, Taoyuan 333, Taiwan; mingfengliao@hotmail.com (M.-F.L.); tulu@ms36.hinet.net (J.-L.H.); lysander@cgmh.org.tw (C.-H.L.); mr5894@cgmh.org.tw (M.-Y.C.); 2Department of Life Science, School of Life Science, National Taiwan Normal University, Taipei 106, Taiwan; ktlu@ntnu.edu.tw; 3Department of Neurology, New Taipei Municipal TuCheng Hospital, Chang Gung Memorial Hospital, Chang Gung University, New Taipei City 236, Taiwan; 4Graduate Institute of Mind, Brain and Consciousness, Taipei Medical University, Taipei 110, Taiwan; 5Brain and Consciousness Research Center, Shuang Ho Hospital, New Taipei City 235, Taiwan

**Keywords:** neuropathic pain, autophagy, apoptosis, proinflammatory cytokines

## Abstract

Neuropathic pain indicates pain caused by damage to the somatosensory system and is difficult to manage and treat. A new treatment strategy urgently needs to be developed. Both autophagy and apoptosis are critical adaptive mechanisms when neurons encounter stress or damage. Recent studies have shown that, after nerve damage, both autophagic and apoptotic activities in the injured nerve, dorsal root ganglia, and spinal dorsal horn change over time. Many studies have shown that upregulated autophagic activities may help myelin clearance, promote nerve regeneration, and attenuate pain behavior. On the other hand, there is no direct evidence that the inhibition of apoptotic activities in the injured neurons can attenuate pain behavior. Most studies have only shown that agents can simultaneously attenuate pain behavior and inhibit apoptotic activities in the injured dorsal root ganglia. Autophagy and apoptosis can crosstalk with each other through various proteins and proinflammatory cytokine expressions. Proinflammatory cytokines can promote both autophagic/apoptotic activities and neuropathic pain formation, whereas autophagy can inhibit proinflammatory cytokine activities and further attenuate pain behaviors. Thus, agents that can enhance autophagic activities but suppress apoptotic activities on the injured nerve and dorsal root ganglia can treat neuropathic pain. Here, we summarized the evolving changes in apoptotic and autophagic activities in the injured nerve, dorsal root ganglia, spinal cord, and brain after nerve damage. This review may help in further understanding the treatment strategy for neuropathic pain during nerve injury by modulating apoptotic/autophagic activities and proinflammatory cytokines in the nervous system.

## 1. Introduction

Neuropathic pain is induced by damage to the somatosensory nervous system [1]. Neuropathic pain is difficult to manage, and epidemiological studies have shown that the prevalence of neuropathic pain ranges from 3 to 17% [2,3,4]. Current treatments of neuropathic pain include tricyclic antidepressants (TCAs), serotonin–noradrenaline reuptake inhibitors (SNRIs), and the α2δ subunit of voltage-dependent calcium channel ligands (pregabalin/gabapentin) [3,5]. However, most of these medication treatments only have moderate efficacy based on the number needed to treat (NNT), and pharmacological treatments for neuropathic pain are effective in <50% of patients, according to previous reviews [3,4,5]. A new strategy to treat neuropathic pain is urgent and worth developing.

Autophagy and apoptosis are two crucial processes when cells encounter damage or stress [6,7]. Autophagy is a self-digestion process involved in protein and organelle degradation to improve the survival rate of cells in a stressful environment [6]. Apoptosis is a kind of programmed cell death and cell shrinkage when cells encounter stress or damage [7]. Apoptosis and autophagy both play an essential role in cancer, immunity, and neurodegenerative diseases [6,8,9,10,11]. Furthermore, autophagy [12,13] and apoptosis [14,15,16,17,18,19,20,21,22] are also both involved in nerve damage and the process of neuropathic pain formation. It is easy to predict that injured neurons will have apoptotic changes after nerve damage. Some animal studies have demonstrated increased apoptotic changes in the injured nerve, the dorsal root ganglia [14,15,16,17,18], and spinal cord [19,20,21,22] after peripheral nerve injury. Studies have found that different interventions, including medicines and hyperbaric oxygen therapy, can simultaneously attenuate neuropathic pain and suppress apoptotic activities in the dorsal root ganglia or spinal cord [14,15,19,20,21,22]. However, there is no direct evidence that inhibition of apoptotic activities in the injured neurons can attenuate pain behavior. In contrast to apoptosis, many studies with different animal models have shown changed autophagic activities in the injured nerve [23,24,25,26,27,28,29], spinal cord [30,31,32,33,34,35,36,37,38,39,40,41,42], and even brain [43] after nerve injury. These studies also showed that upregulated autophagic activities can directly alleviate neuropathic pain [24,26,27,32,33,37,38]. Additionally, autophagy and apoptosis have crosstalk through various proteins [44,45] and are both regulated by proinflammatory cytokines [46,47,48,49,50,51]. Studies have shown that proinflammatory cytokines play an essential role in promoting apoptotic activities [50,51] and neuropathic pain formation [52,53], while autophagy can inhibit proinflammatory cytokine activities [46,47,48,49]. Thus, agents that can upregulate autophagic activities and suppress proinflammatory cytokines activities have the potential to treat neuropathic pain. Liu’s review summarized the complex proteins and molecular modifications of the autophagy process in neuropathic pain conditions [13]. However, there has been no systematic review of the role of apoptosis and the interaction between autophagy, apoptosis, and proinflammatory cytokines in neuropathic pain formation. In this review, we will discuss autophagic/apoptotic activity changes and their interactions with proinflammatory cytokines based on the different anatomical locations of the nervous system, including the injured nerve, Schwann cells, dorsal root ganglion, spinal cord, and brain. We will also summarize the alteration of pain behaviors after modulating autophagic/apoptotic activities.

## 2. Autophagy in Neuropathic Pain Formation

There are three types of autophagy: macroautophagy, microautophagy, and chaperone-mediated autophagy (CMA) [6,8]. Different animal pain models have demonstrated alterations in macroautophagic activities in the injured nerve, dorsal root ganglia [23,24,25,26,27,28,29], spinal cord [30,31,32,33,34,35,36,37,38,39,40,41,42], and brain [43] after peripheral nerve damage. At the beginning of the macroautophagic process, Beclin-1 and Ambra1 (both are the upper stream proteins of the macroautophagy process) at the endoplasmic reticulum respond to the stress signaling pathway and initiate phagophore formation. In the next step, several proteins, including autophagy-related protein 5 (Atg5)–Atg12, Atg16L complex, Atg7, microtubule-associated protein 1A/1B-light chain 3 (LC3)-II, and p62, are involved in autophagosome formation [6,10,54,55]. Then, autophagosomes fuse with lysosomes, leading to autolysosome formation and further degradation [6,10]. Chloroquine, which blocks the fusion between autophagosomes and lysosomes, can be the autophagy inhibitor in the late stage of autophagy [56]. Another important autophagy regulatory pathway includes a class I phosphatidylinositol 3-kinase (PI3K) and a downstream mammalian target for rapamycin (mTOR) that inhibits the autophagic process [6,8]. Rapamycin, which inhibits mTOR activities, can upregulate autophagic activities, while 3-methyladenine (3-MA), which inhibits PI3K activities, can downregulate autophagic activities [6,56]. Microautophagy needs to be studied by electron microscopy and is still obscure in mammalian cells [57]. On the other hand, CMA is only observed in mammalian cells and has protein markers, including heat shock-cognate chaperone of 70 kDa (HSC70) and lysosome-associated membrane protein type-2A (LAMP2A) [57,58]. Both macroautophagy and CMA play essential roles in central neurological degeneration diseases [59]. The CMA pathway compensates when the macroautophagy process is inhibited; in contrast, CMA blockage induces compensative upregulation of macroautophagy [59].

In peripheral nerve injury, only a few studies discussed the role of CMA in the damaged nerve. LAMP2A expressions increased on the sciatic nerve of rats with chronic inflammatory demyelinating polyneuropathy (CIDP) [60]. Many studies focused on macroautophagy changes in the neurons of animals after nerve damage with rapamycin or 3-MA treatment [23,24,25,26,27,28,29,30,31,32,33,34,35,36,37,38,39,40,41,42]. Most studies proposed that when macroautophagic activities increase, the expression levels of LC3-II, Bclin-1, and Atg5-Atg12 increase, but p62 levels (delivering ubiquitinated cargoes to autophagosomes) decrease [23,24,25,26,27,28,29,30,31,32,33,34,36,37,38,39,40,41,42]. However, some authors stated that, lately, increased LC3-II expressions indicate that the autophagic flux is blocked and autophagic activities are decreased [35]. Herein, we summarized studies and discussed the alteration of autophagic activities at different anatomical locations in various animal pain models.

### 2.1. Autophagic Activity Changes in Injured Nerves, including Schwann Cells and Dorsal Root Ganglia, after Nerve Injury

It is easy to understand that injured nerves, including Schwann cells and dorsal root ganglia, which undergo stress and damage, show autophagic activity changes to adapt to peripheral nerve damage. Elevated autophagic activity may help myelin debris clearance, which is possibly related to chronic pain modulation. Many studies that used different animal models (including chronic constriction injury (CCI), sciatic nerve transection (SNT), spinal nerve ligation (SNL), and sciatic nerve crush injury (SNC)) have shown consistent results that the elevated autophagic activities are measured by different autophagic proteins (Beclin-1, Atg5, Atg7, LC3II) in the injured nerve axons, Schwann cells, and dorsal root ganglia from 6 h to 21 days after nerve injury [23,24,25,26,27,28,29]. Furthermore, rapamycin can upregulate the autophagic activities in the dorsal root ganglia and Schwann cells and suppress pain behavior. On the other hand, 3-MA or transgenic methods that downregulate autophagic activities in the dorsal root ganglia and Schwann cells can enhance pain behavior [24,25,26,27,28]. Table 1 summarizes the findings of those studies.

### 2.2. Autophagic Activity Changes in the Spinal Cord after Nerve Injury

The spinal dorsal horn involved in central sensitization is another critical area in neuropathic pain formation [53,61,62]. Proinflammatory cytokines activated in the spinal dorsal horn (SDH) after peripheral nerve damage promote neuropathic pain formation [52,53]. Many studies have discussed the relations between proinflammatory cytokines and autophagic activity changes in the SDH in neuropathic pain by different animal models. Researchers studied the pain behavior and autophagic protein expressions in the spinal cord after direct administration of rapamycin and 3-MA [32,37,38,39]. Other studies utilized indirect methods; researchers monitored the effects of different proinflammatory mediators on pain behavior and autophagic protein expression in the spinal dorsal horn [34,35,36,40,41,42]. Thus, these findings hypothesized that autophagic activity could affect neuropathic pain indirectly.

Unlike the studies of dorsal root ganglia and injured nerves, surveys of autophagic expression changes in the spinal dorsal horn have shown inconsistent results. Some studies showed upregulated autophagic activities in the different types of cells in the spinal cord after nerve injury [30,31,32,33,39,40,41,42]; other studies demonstrated the opposite results [34,35,36,37,38]. Although those studies did not have consistent results for the autophagic activity changes in the spinal cord after nerve injury, most studies showed that upregulation of autophagy could suppress pain behavior [30,31,32,33,34,35,36,37,38]. These inconsistent results are probably caused by different animal models and study designs. Below, we summarized those studies according to their findings in Table 2.

#### 2.2.1. Increased Autophagic Activities in the Spinal Cord Neurons, Decreased Autophagic Activities in the Spinal Cord Microglia and Astrocytes after Nerve Injury, and Autophagy Acts as a Pain Suppressor

Several studies from neuropathic pain models (including CCI, SNL, spared nerve injury (SNI)) and diabetic rats had similar results, showing that autophagic activities (LC3-II and Becline-1 expressions) on the spinal cord (mainly on neurons) were elevated from day 1 to day 21 after nerve injury. Furthermore, autophagy inhibition by chloroquine and the PI3K inhibitor can enhance pain behavior, while autophagy upregulation by rapamycin can suppress pain behavior [30,31,32,33]. In contrast to previous results that showed that autophagic activities on the spinal neurons were upregulated in neuropathic pain, other research showed that autophagic activities on the spinal cord decreased (mainly on microglia and astrocytes) from day 2 to day 28 after nerve injury [34,35,36,37,38]. Furthermore, those studies demonstrated that autophagic upregulation could suppress pain behavior, possibly by suppressing inflammatory responses [34,35,36,37,38]. However, many of those studies did not observe the direct effects of autophagy inducers and autophagy suppressors. Instead, they followed the impact of different anti-inflammatory agents (e.g., microRNA-195 inhibitor, Koumine, resveratrol) on pain behavior and autophagic activities on the spinal cord and then hypothesized the analgesic effects of autophagic upregulation on neuropathic pain [34,35,36,37,38].

#### 2.2.2. Increased Autophagic Activities in the Spinal Cord Neurons and Microglia after Nerve Injury, and Autophagy Acts as a Pain Enhancer

However, a few studies that used CCI and SNL models showed that autophagy might act as a pain enhancer [39,40,41,42]. A study showed that LC3 expression (co-stained with NeuN) elevated on the SDH of rats with spinal nerve ligation (SNL) and direct intrathecal 3-MA could decrease the pain behavior of rats with SNL [39]. Other studies did not observe the direct effects of autophagy inducers and autophagy suppressors; they studied the pain behavior, proinflammatory cytokine, and autophagic activity changes after administration of different proinflammatory and anti-inflammatory agents (e.g., modified citrus pectin, a TLR-3 agonist, si-ciRS-7) [40,41,42]. They found that autophagy inducers can reverse the anti-inflammatory agent effects and hypothesized that autophagy acts as a pain inducer and has proinflammatory effects [40,41,42].

### 2.3. Autophagic Activity Changes in the Brain after Nerve Injury

In addition to the dorsal root ganglia and spinal dorsal horn, different brain areas, including the anterior cingulate cortex, insula, amygdala, and nucleus accumbens, are also involved in neuropathic pain perception formation [53,63,64]. Neural cellular circuits and molecular components change in those brain areas when neuropathic pain forms [63,64]. A study showed that LC3/GFAP (astrocyte marker)-positive cells increased in the brain tissue (however, the authors did not mention which brain areas) of rats with CCI [43]. Furthermore, lidocaine treatment attenuated mechanical allodynia and increased LC3/GFAP-positive cells in the brain of rats with CCI [43]. Lidocaine treatment further increased the LC3-II/LC3-I ratio but decreased proinflammatory cytokine expression (IL-1β and TNF-α) in astrocyte cultures that received a lipopolysaccharide (LPS) challenge [43]. 3-MA reversed the effects of lidocaine on astrocyte cultures after the LPS challenge. This study hinted that brain tissue might also exhibit increased autophagic activities under neuropathic conditions. Further upregulated autophagic activities in brain tissue may suppress proinflammatory responses and attenuate neuropathic pain [43].

## 3. Apoptosis in Neuropathic Pain Formation

### 3.1. Apoptotic Activity Changes in the Injured Nerve and Dorsal Root Ganglia after Nerve Injury

The extrinsic apoptotic pathway is triggered by extracellular death ligands (such as TNF-α) and the intrinsic apoptotic pathway mediated by B-cell lymphoma 2 (Bcl-2)-associated X protein (Bax)/Bak on the mitochondrial membrane initiates further apoptosis processes. Subsequently, cytochrome c is released from mitochondria and activates the final caspase-3 cascades of apoptosis [7]. It is reasonable to predict that apoptotic activities in injured nerve and dorsal root ganglia increase after nerve injury. Several studies have demonstrated that apoptotic activities in injured dorsal root ganglia increased after injury and some agents could simultaneously suppress pain behavior and apoptotic activities [14,15,16,17,18]. Most studies measured the levels of antiapoptotic proteins (Bcl-2) and proapoptotic proteins (Bax, caspase-3, caspase-7, caspase-9, cytochrome-c) involved in the apoptotic pathway [7] and the activities of terminal deoxynucleotidyl transferase dUTP nick end labeling (TUNEL) assays, which can detect apoptotic deoxyribonucleic acid (DNA) fragmentation [65] to indicate apoptotic activities in dorsal root ganglia.

Several studies that used CCI, SNC, and SNT models demonstrated consistent results that apoptotic activities (measured by cleaved caspase-3, cytochrome c, and the TUNEL assay) on dorsal root ganglia neurons and satellite cells increased from day 2 to day 28 after nerve injury [14,15,16,17,18]. They also demonstrated that different agents (including ALCAR, estradiol, and recombinant human erythropoietin) could suppress pain behavior and decrease apoptotic activities in injured dorsal root ganglia at the same time [14,15,16,17,18]. However, there is no direct evidence that agents showing inhibition of apoptotic activities in the injured neurons could attenuate pain behavior. Those results are summarized in Table 3.

### 3.2. Apoptotic Activity Changes in the Spinal Cord after Nerve Injury

In contrast to the dorsal root ganglia, different studies have shown inconsistent results for the apoptotic changes in the spinal dorsal horn in neuropathic pain. Studies that used SNC and CCI models demonstrated increased apoptotic activities in the dorsal root ganglia but not in the spinal cord after nerve injury [14,17]. Another study that used the SNI model demonstrated similar results; the TUNEL assay and cleaved caspase-3 activities by immunohistochemical studies were not revealed in the neurons of the dorsal spinal cord one week after nerve injury. However, they found that TUNEL activities were co-stained with Iba1 (a microglial marker) [66].

On the other hand, other studies that used CCI models and mice with vincristine showed different results [19,20,21,22]. They demonstrated increased apoptotic activities (measured by Bax, apoptotic protease-activating factor-1 (apaf-1), caspase-3, caspase-9, cytochrome c, Bax, and TUNEL activities) in the spinal dorsal horn (mainly neurons) from day 3 to day 16 after nerve injury. Furthermore, those studies showed that different methods, including antioxidants, reactive oxygen species (ROS) scavengers, mitoquinone, and hyperbaric oxygen (HBO) therapy, can suppress pain behavior and decrease apoptotic activities in the spinal dorsal horn at the same time [19,20,21,22]. However, there is no direct evidence demonstrating that agents inhibiting apoptotic activities in the spinal dorsal horn could attenuate pain behavior. Table 4 summarizes the findings of those studies.

## 4. Relationships between Autophagy and Apoptosis in Neuropathic Pain Formation

Autophagy and apoptosis have crosstalk through various proteins, including Atg5–Atg12, Beclin-1/Bcl-2, and caspase-3 under stress conditions [44,45]. Additionally, a non-classical autophagy process called LC3-associated phagocytosis (LAP) can help remove apoptotic cells by macrophages and inhibit proinflammatory processes [67,68]. In addition, autophagy and apoptosis are both regulated by proinflammatory cytokines [46,47,48,49,50,51]. Autophagy has strong interactions with proinflammatory and anti-inflammatory cytokines, mainly in immune cells, including lymphocytes and macrophages [46,47,48,49]. Proinflammatory cytokines, including IFN-γ, TNF-α, IL-1β, and IL-6, have been shown to have the ability to activate autophagic processes, while anti-inflammatory cytokines, including IL-4 and IL-10, could inhibit autophagic activities [47,48,49]. Comparatively, autophagy could suppress the activities of proinflammatory cytokines, mainly the IL-1 family, but also IL-6, TNF-α, and IL-17 [46,47,48,49]. TNF-α itself can activate the extrinsic apoptosis pathway [7]. Proinflammatory cytokines could directly activate apoptosis in non-neuronal cells, including β-cells and epithelial cells [50,51]. The contributions of proinflammatory cytokines, including IL-1 and TNF-α, to apoptosis in neurodegenerative diseases such as Alzheimer’s disease and amyotrophic lateral sclerosis are also well established [11,69]. In addition to autophagy and apoptosis, proinflammatory cytokines play important roles in neuropathic pain modulation, and inhibiting proinflammatory cytokines may help to attenuate neuropathic pain [52,53]. Therefore, proinflammatory cytokines could be the key regulators between autophagy and apoptosis in neuropathic pain formation [46,47,48,49,50,51,52,53] (Figure 1).

Several studies have focused on autophagic and apoptotic activity changes in damaged nerves simultaneously and discussed the relationship between autophagy and apoptosis after nerve damage. A study that used the SNC model showed that rats with SNC had higher autophagic activities (LC3-II/LC3-1 ratios and autophagosome numbers) and apoptotic activities (TUNEL assays) in the injured nerve than sham rats one week after nerve injury [27]. Furthermore, rapamycin treatment (for five continuous days after nerve injury) improved rats’ motor functions from one to two weeks after nerve injury, upregulated autophagic activities on the injured nerve, and promoted nerve regeneration (measured by Neurofilament 200 [NF200] and myelin basic protein [MBP] density) one week after nerve injury. Rapamycin treatment also decreased TUNEL assay activities in the injured nerve one week after nerve injury. Conversely, 3-MA had the opposite effect. This study demonstrated that autophagy upregulation could possibly inhibit apoptotic activities in the injured nerve. However, there were no further explanations of the detailed mechanisms in their study [27]. Another study that used the CCI model demonstrated that Koumine treatment could suppress the pain behavior of rats with CCI, downregulate proinflammatory cytokine (IL-1β and TNF-α) expression, and upregulate autophagic activities in the spinal cord. Autophagy inhibitors (chloroquine) can reverse these effects [35]. In this study, Koumine treatment also suppressed apoptotic activities (cleaved caspase 3, Bcl-2/Bax) in the spinal cord of rats with CCI at the same time (day 9 after nerve injury). The authors suggested that the antiapoptotic abilities of Koumine were probably mediated by suppressing proinflammatory cytokine expressions [35]. Our recent studies demonstrated similar findings [29]. We found that granulocyte-colony-stimulating factor (G-CSF) treatment can suppress pain behavior, upregulate autophagic activities (at the early phase after nerve injury), suppress proinflammatory cytokine expression, and downregulate apoptotic activities (at the late phase after nerve injury) in the injured nerve and dorsal root ganglia. We emphasized that upregulated autophagic activities developed first, followed by suppression of proinflammatory cytokine expression, and, furthermore, demonstrated the attenuation of apoptotic activities last, after G-CSF treatment [29]. Another recent study that used the SCI model showed curcumin (a traditional antibacterial agent) promoted motor recovery in rats with SCI, reduced apoptotic activities, suppressed proinflammatory cytokine expressions, and enhanced autophagic activities in the spinal cord of rats with SCI. Furthermore, chloroquine (an autophagy inhibitor) eliminated the protective effect of curcumin, including the suppression of proinflammatory cytokines in rats with SCI [70]. Previous studies, including ours, demonstrated the interactions of autophagy, apoptosis, and proinflammatory cytokine expression when neurons encounter stress and damage. Medications such as Koumine, G-CSF, and curcumin, which could suppress proinflammatory cytokines and apoptotic activities and enhance autophagic activities on the injured neurons, have the potential to treat neuropathic pain.

## 5. Conclusions and Perspectives on This Review

Most studies have shown consistent findings, suggesting that autophagic and apoptotic activities in injured nerves and dorsal root ganglia increase after nerve damage. Evidence has shown that the upregulated autophagic activities on the injured nerve and dorsal root ganglia after nerve damage could possibly help myelin clearance and promote nerve regeneration, which can attenuate pain behavior. However, there is less convincing evidence demonstrating that the pain behavior could be suppressed by agents that directly inhibit apoptosis. On the other hand, the autophagic and apoptotic changes in the spinal cord after peripheral nerve injury have shown inconsistent results in different studies. Complex central sensitization mechanisms in the spinal cord, different animal models, different study designs and times, and the use of various treatment agents can cause these inconsistent findings. The crosstalk or interaction of autophagy and apoptosis in spinal cord neuropathic pain formation mainly involves the modulation of proinflammatory cytokines. Alterations in autophagic/apoptotic and proinflammatory cytokine activities over time in the spinal dorsal horn play essential roles in neuropathic pain formation after nerve damage. The alterations of autophagic/apoptotic activities and their interactions with proinflammatory agents in the injured nerve, dorsal root ganglia, spinal cord, and brain are summarized in Figure 2. Further detailed studies of autophagy, apoptosis, proinflammatory cytokines, and their interactions in neuropathic pain formation in the spinal cord after nerve injury are warranted to develop a new strategy to treat neuropathic pain.

## Figures and Tables

**Figure 1 ijms-23-02685-f001:**
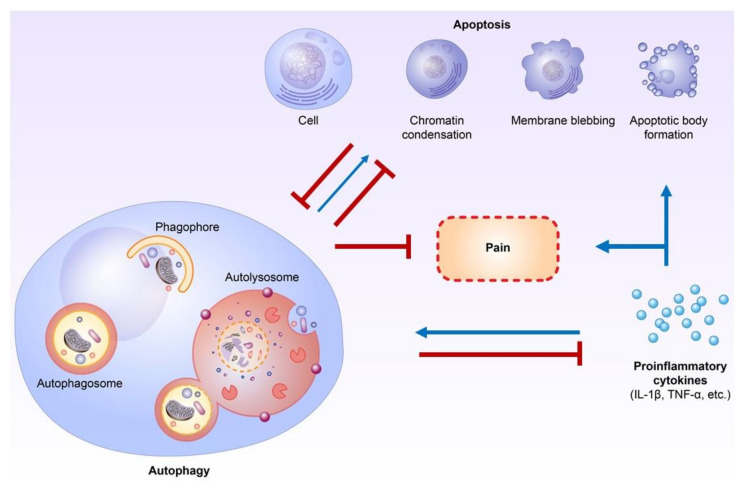
Crosstalk between autophagy, apoptosis, and proinflammatory cytokines during neuropathic pain formation. Autophagy plays an important role in inflammation and proinflammatory cytokine regulation. Furthermore, proinflammatory cytokines directly influence apoptosis in non-neuronal cells and probably neurons of the central nervous system. The crosstalk or interaction of autophagy and apoptosis in neuropathic pain formation occurs mainly through the modulation of proinflammatory cytokines. (Blue lines: activation; Red lines: inhibition).

**Figure 2 ijms-23-02685-f002:**
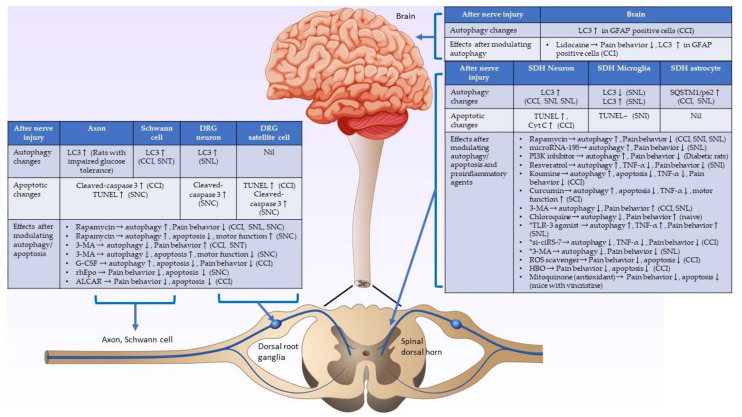
Schematic diagram showing the underlying mechanisms of autophagic and apoptotic activities in the nerve injury. Summary of autophagic/apoptotic activity changes in the injured axon, Schwann cell, spinal dorsal horn, and brain after nerve injury, and the effects after modulating autophagic/apoptotic and proinflammatory cytokines activities after nerve injury. (CCI: chronic constriction injury, SCI: spinal cord injury, SNC: sciatic nerve crush injury, SNI: spared nerve injury, SNL: spinal nerve ligation, ↑: activities increase, ↓: activities decrease, -: activities do not change. *: studies showed that autophagy is a pain inducer.).

**Table 1 ijms-23-02685-t001:** Summary of the alterations of autophagic activities in the injured nerve, including Schwann cells and dorsal root ganglia, after nerve injury in different models of rats and mice.

Reference	Animal Models	Western Blot Findings	IHC Findings	Effects of Therapeutic Agents
Kosacka et al., 2013 [23]	WOKW rats (hyperinsulinemia and impaired glucose tolerance).	Atg 5, Atg7, LC3-II/LC3-I ratio in the sciatic nerve of WOKW rats increased.	Autophagic proteins (LC3) were co-stained with S100 (Schwann cell), PGP9.5 (nerve fiber), and Iba1 (macrophage).	Nil.
Marinelli et al., 2014 [24]	*Ambra1* and GFP-LC3 transgenic mice with chronic constriction injury (CCI).	Rapamycin increased the LC3-II/LC3-1 ratio in the injured sciatic nerve; 3-MA decreased LC3-II/LC3-1 ratio on the injured nerve.	Autophagic proteins (LC3) were co-stained with GFAP (Schwann cell).	Rapamycin intraplantar (i.pl.) injected 3 days after CCI suppressed pain behavior; 3-MA enhanced pain behavior.
Liao et al., 2021 [29]	Rats with chronic constriction injury (CCI).	LC3-II in the injured sciatic nerve and dorsal root ganglia increased from day 3 to day 7 after nerve injury.	Nil.	Granulocyte-colony-stimulating factor (G-CSF) suppressed pain behavior and upregulated autophagic activities.
Huang et al., 2016 [27]	Rats with a sciatic nerve crush injury (SNC).	LC3-II/LC3-1 ratio and the number of autophagosomes in the injured sciatic nerve increased 1 week after nerve injury.	Nil.	Intraperitoneal injections of rapamycin improved motor function and upregulated autophagic activities.
Jang et al., 2016 [28]	*Atg 7* knockout mice with a sciatic nerve crush injury (SNC).	LC3-II at the distal stump of the injured sciatic nerve increased from 6 h to 48 h following nerve injury.	Nil.	Pharmacological intervention of lysosomal function or inhibition of autophagy via Schwann cell-specific knockout of the *Atg 7* gene caused a significant delay in myelin clearance.
Guo et al., 2015 [26]	Mice with L5 spinal nerve ligation (SNL).	LC3-II in the injured dorsal root ganglia of mice with SNL increased from day 3 to day 21 after nerve injury.	Increased LC3 expressions were found on the DRG neurons.	Injection of rapamycin into DRG upregulated autophagic activities and suppressed pain behavior.
Gomez-Sanchez et al., 2015 [25]	*Atg 7fl/fl* and GFP-LC3 transgenic mice with sciatic nerve transection (SNT).	LC3-II, Beclin-1, Atg5-Atg12, Atg7, Atg 16L1 in the injured nerve increased from day 2 to day 14 after nerve injury. LC3-II had the highest expressions on day 2 after nerve injury.	Autophagic proteins (LC3) were co-stained with MPZ (myelin).	Pharmacological (3-MA) and genetic inhibition (*Atg7fl/fl* mice) of autophagy impaired myelin clearance.

**Table 2 ijms-23-02685-t002:** Summary of the alterations of autophagic activities in the spinal cord after nerve injury in different models of rats and mice.

Reference	Animal Models	Western Blot Findings	IHC Findings	Effects of Therapeutic Agents
Increased autophagic activities in the spinal cord neurons, decreased autophagic activities in the spinal cord microglia and astrocytes after nerve injury, and autophagy acts as a pain suppressor.				
Berliocchi et al., 2011 [30], 2015 [31]	Mice with chronic constriction injury (CCI), spared nerve injury (SNI), and spinal nerve ligation (SNL).	LC3-II and p62 increased in the mice with spinal nerve ligation (SNL) on day 7 after nerve injury. LC3-II increased in the mice with spared nerve injury (SNI) on day 14 after nerve injury. Beclin-1 increased in the mice with chronic constriction injury (CCI) on day 14 after nerve injury.	Autophagic proteins (p62) were co-stained with NeuN in the spinal cord.	Intrathecal chloroquine (can inhibit autophagic flux and block the late stage of autophagy) enhanced pain behavior.
Chen et al., 2018 [32]	Rats with chronic constriction injury (CCI).	LC3-II and Beclin-1 increased from day 1 to day 7 (peak at day 3) after nerve injury. p62 decreased from day 1 to day 7 after nerve injury.	Nil.	Intraperitoneal rapamycin before chronic constriction injury suppressed pain behavior, upregulated LC3-II/Beclin-1 expressions, and suppressed astrocyte activation in the spinal cord. 3-MA showed the opposite effect.
Liu et al., 2020 [33]	Rats received streptozotocin (STZ) injection (diabetic rats).	p-PI3K, p-AKT, and p-mTOR decreased, but LC3-II and Beclin1 increased 3 weeks after STZ injection.	Nil.	Intravenous PI3K inhibitor (LY294002) suppressed pain behavior and increased LC3-II expressions in the spinal cord of diabetic rats.
Jin et al., 2018 [35]	Rats with chronic constriction injury (CCI).	LC3-II/I ratio and p62 increased on day 9 after nerve injury. (The authors suggested that increased LC3-II indicated that autophagic flux was blocked in the late stage of autophagy and concluded that autophagic activities in the spinal cord decreased after nerve injury.)	Autophagic proteins (LC3) were co-stained with GFAP in the spinal cord.	Subcutaneously (s.c.) Koumine treatment suppressed pain behavior, upregulated autophagic activities, and downregulated proinflammatory cytokine expressions in the spinal cord of rats with CCI. Intrathecal chloroquine (autophagy inhibitor) blocked the effects of Koumine.
Wang et al., 2020 [36]	Rats with spared nerve injury (SNI).	LC3-II decreased but triggering receptor expressed on myeloid cells 2 (TREM2), p62, and proinflammatory cytokine increased on day 7 after nerve injury.	Nil.	Intrathecal resveratrol treatment (suppressed TREM2 expressions) suppressed pain behavior, suppressed proinflammatory cytokine, and increased LC3-II expressions in the spinal cord. 3-MA (autophagy suppressor) reduced the analgesic effects of resveratrol.
Hu et al., 2021 [37]	Rats with spared nerve injury (SNI).	LC3-II decreased, but Beclin-1 and p62 increased from day 7 to day 14 after nerve injury.	Nil.	Intravenous rapamycin treatment decreased pain behavior, increased LC3-II expressions and autophagosome number in the spinal cord, and suppressed C- and A-fiber-evoked field potentials from day 7 to day 14 after nerve injury.
Shi et al., 2013 [34]	Rats with spinal nerve ligation (SNL).	LC3-II/LC3-1 ratio of the primary microglia culture isolated from rats with SNL decreased from day 2 to day 14 after nerve injury. P62 of the primary microglia culture isolated from rats with SNL increased from day 5 to day 14 after nerve injury.	Autophagic proteins (LC3-II) were co-stained with Iba1 in primary microglia culture.	Intrathecal administration of microRNA-195 inhibitor reduced the pain behavior of rats with SNL and increased the LC3-II/LC3-I ratio in the spinal dorsal horn.
Li et al., 2021 [38]	Mice with spinal nerve ligation (SNL).	LC3-II/LC3-I ratio and Atg5 expressions decreased, but SQSTM1/p62 (autophagy receptor) increased from day 7 to day 28 after nerve injury.	SQSTM1 was mainly co-stained with GFAP rather than NeuN. Electron microscopic studies showed the number of autophagosomes decreased, mainly in the astrocyte of mice with SNL.	Intrathecal rapamycin treatment on day 7 to day 9 after nerve injury suppressed pain behavior on day 10 and day 14 after nerve injury and increased LC3-II/LC3-I ratio in the spinal cord. 3-MA showed the opposite effects.
Increased autophagic activities in the spinal cord neurons and microglia after nerve injury, and autophagy acts as a pain enhancer.				
Cai et al., 2020 [42]	Rats with chronic constriction injury (CCI).	LC3-II, circular RNAs-7 (ciRS-7), and proinflammatory cytokines (IL-6, IL-12, TNF-α) levels increased from day 7 to day 20 after nerve injury.	Nil.	Intrathecal si-ciRS-7 treatment suppressed pain behaviors and decreased autophagic proteins (LC3-II) and proinflammatory cytokines (IL-12, TNF-α) expressions.
Zhang et al., 2013 [39]	Rats with spinal nerve ligation (SNL).	Nil.	Autophagic proteins (LC3 and Beclin-1) were co-stained with NeuN and Calretinin (a marker of GABAergic interneurons) in the spinal dorsal horn and increased on day 14 after nerve injury.	Intrathecal 3-MA 3 days after SNL decreased pain behavior from day 7 to day 10 after nerve injury.
Ma et al., 2016 [40]	Rats with spinal nerve ligation (SNL).	LC3-II of the primary microglia culture isolated from the rats that received SNL increased on day 10 after nerve injury.	Nil.	Intrathecal administration of modified citrus pectin (a kind of anti-inflammatory protein) suppressed pain behavior of rats with SNL, and rapamycin reversed those effects.
Weijia Chen et al., 2017 [41]	Rats with spinal nerve ligation (SNL).	LC3-II of the primary microglia culture isolated from the rats with SNL increased but p62 levels decreased on day 10 after nerve injury.	Autophagic proteins (LC3) were co-stained with Iba-1 (microglia marker) in the spinal cord.	Intrathecal TLR-3 agonist enhanced pain behavior of rats with SNL and increased autophagy and proinflammatory cytokines (IL-1β and TNF-α) expressions in the dorsal horn of rats with SNL. Intrathecal 3-MA administration reversed previous findings.

**Table 3 ijms-23-02685-t003:** Summary of the alterations of apoptotic activities in the injured nerve and dorsal root ganglia after nerve injury in different rat models.

Reference	Animal Models	Apoptotic Activities	IHC Findings	Effects of Therapeutic Agents
Mannelli et al., 2009 [15]	Rats with chronic constriction injury (CCI).	Cleaved caspase 3, cytochrome c, DNA fragmentation levels in the injured nerve increased on day 15 after nerve injury.	Nil.	Intraperitoneal ALCAR twice per day for 15 days suppressed pan behavior and decreased cleaved-caspase 3, cytochrome c, and DNA fragmentation levels in the injured nerve.
Schaeffer et al., 2010 [17]	Rats with chronic constriction injury (CCI).	TUNEL assay activities in the dorsal root ganglia increased on day 30 (not day 5 and day 15) after nerve injury.	TUNEL activities were co-stained with anti-glutamine synthetase (satellite cells marker). Aromatase (estradiol-synthesizing enzyme) was co-stained with NeuN but not anti-glutamine synthetase (satellite cells marker)	Letrozole, which blocked aromatase activities, increased apoptotic activities in the dorsal root ganglia of rats with CCI.
Campana et al., 2003 [14]	Rats with spinal nerve crush (SNC) injury.	TUNEL assay activities in the dorsal root ganglia increased on day 2 after nerve injury.	Nil	Subcutaneous administration of recombinant human erythropoietin (rhEpo) one day before nerve injury suppressed pain behavior and apoptotic activities in the dorsal root ganglia from day 2 to day 14 after nerve injury.
Sekiguchi et al., 2009 [16]	Rats with spinal nerve crush (SNC) injury.	In situ Oligo labeling (ISOL) in the dorsal root ganglia increased from day 2 to day 28 after nerve injury.	Cleaved-caspase 3 on dorsal root ganglia were co-stained with NeuN and GFAP (satellite cells marker).	Nil.
Wiberg et al., 2018 [18]	Rats with sciatic nerve transection (SNT).	Caspase-3, caspase-8, caspase-12, caspase-7, and calpain expressions in the dorsal root anglia increased on day 7, day 14, and day 28 after nerve injury.	Nil.	Nil.

**Table 4 ijms-23-02685-t004:** Summary of the alterations of apoptotic activities in the spinal dorsal horn in different models of rats and mice.

Reference	Animal Models	Apoptotic Activities	IHC Findings	Effects of Therapeutic Agents (Inhibit Apoptosis)
Schaeffer et al., 2010 [17]	Rats with chronic constriction injury (CCI).	TUNEL assay activities in the spinal cord did not increase on days 5, 15, and 30 after nerve injury.	Nil.	Nil.
Campana et al., 2003 [14]	Rats with spinal nerve crush (SNC) injury.	TUNEL assay activities in the spinal cord did not increase on day 2 after nerve injury.	Nil.	Subcutaneous administration of recombinant human erythropoietin (rhEpo) 1 day before nerve injury reduced pain behavior from day 2 to day 14 after nerve injury.
Siniscalco et al., 2007 [19]	Mice with chronic constriction injury (CCI).	Bax, apoptotic protease-activating factor-1 (apaf-1), caspase-9 mRNA expressions, and TUNEL and caspase-3 activities increased on day 3 after nerve injury.	TUNEL activities were co-stained with NeuN on day 3 after nerve injury.	Intraperitoneal phenyl-N-tert-butylnitrone (ROS scavenger) suppressed pain behavior from day 1 to day 3 after nerve injury and decreased apoptotic activities in the spinal cord on day 3 after nerve injury.
Hu et al., 2015 [20]	Rats with chronic constriction injury (CCI).	TNF-α and caspase-3 mRNA expressions and TUINEL activities increased on day 3 and day 7 after nerve injury, respectively.	Nil.	HBO suppressed pain behavior and apoptotic activities in the spinal cord of rats with CCI on day 3 and day 7 after nerve injury.
Fu et al., 2017 [21]	Rats with chronic constriction injury (CCI).	The number of cleaved caspase-3 and cytochrome C positive neurons increased on day 8 and day 14 after nerve injury.	Cytochrome C was co-stained with NeuN (neuron marker).	Daily HBO therapy suppressed pain behavior and apoptotic activities in the spinal dorsal horn of rats with CCI.
Polga’r et al., 2005 [66]	Mice with spared nerve injury (SNI).	TUNEL assay and cleaved caspase-3 activities did not demonstrate in the apoptotic neurons in the dorsal spinal cord 1 week after nerve injury.	TUNEL activities were co-stained with Iba1 (microglia marker).	Nil.
Chen et al., 2020 [22]	Mice with neuropathic pain induced by vincristine.	Cleaved caspase 3, Bax, and cytochrome c (Cyt-c) increased on day 9 after five consecutive vincristine administrations.	Nil.	Mitoquinone (antioxidant) treatment after vincristine injury suppressed pain behavior of mice, ROS production, and apoptotic activities in the spinal cord of mice.

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
