# Peer review of "The Role of Autophagy and Apoptosis in Neuropathic Pain Formation"

_ijms, 2022, doi:10.3390/ijms23052685_

Round 1

Reviewer 1 Report

In this review, the authors Here, summarized the  apoptotic and autophagic activities in the injured nerve, dorsal root ganglia, spinal cord, brain  damage. This may help   to treat neuropathic pain through modulating apoptotic and autophagic activities in the nervous system. 

The topic is interesting, but has major defects:

1-The paragraphs are so long. For example, the paragraph starting from line 46 to 61. Please summarize the long paragraphs in one or two sentences.

2-What is the role of Ambra1 in autophagy? Please describe.

3-In line 104, what 9.5? What is the importance of the paragraph starting from 104 to 106 lines?

4-In line 133, what are autophagy numbers?

5-In line 137, what is allodynic response?

6-The authors mentioned in lines 194-196 that the suppression of apoptosis in dorsal root ganglia can decrease pain!! How you suppress apoptosis, which is a normal physiologic response to injury? How apoptotic cells are cleared? What is the role of autophagy in clearance of apoptotic cells?I recommend to add a paragraph discussing the role of LC3-associated phagocystosis (LAP) in the clearance of apoptotic cells. You can know about it in this article: Eid N, Ito Y. Oxoglaucine alleviates osteoarthritis by activation of autophagy via blockade of Ca2+ influx and TRPV5/calmodulin/CAMK-II pathway. Br J Pharmacol. 2021 Nov 3. doi: 10.1111/bph.15706

7-In figure 2, what is the meaning of double head arrows?

What are the relations of microautophagy  to to your story?The diagram is incorrect. The  autophagosome fuse with what in left side of the figure?  

Author Response

Thanks for your encouragement and constructive suggestions. We believe that your criticisms can significantly improve the quality of our manuscript. We made the point-to-point responses to your questions and revised them in the new version of manuscript.

  1. The paragraphs are so long. For example, the paragraph starting from line 46 to 61. Please summarize the long paragraphs in one or two sentences.

Response: Thanks for your constructive suggestions. We summarized those long paragraphs in two sentences “Apoptosis is a kind of programmed cell death and cell shrinkage when cells encounter stress or damage. Autophagy is a self-digestion process involved in protein and organelle degradation to improve the survival rate of cells in the stressful environment.” We have shortened all redundant sentences in the revised manuscript.

  1. What is the role of Ambra1 in autophagy? Please describe.

Response: Beclin-1 and Ambra1 at the endoplasmic reticulum responds to the stress signaling pathway and initiate phagophore formation in the early phase of autophagy [1]. Both Beclin-1 and Ambra1 are the upper stream proteins of the autophagy process. We have added those descriptions and the references in the beginning of page 3.

  1. In line 104, what 9.5? What is the importance of the paragraph starting from 104 to 106 lines?

Response: To shorten the manuscript, we delete the descriptions of the results of one single study including description of PGP 9.5, a neural marker in the beginning of page 4. We have also deleted paragraphs from 104-106 lines, which are less relevant to the autophagic activity on the injured nerve in that section.  

  1. In line 133, what are autophagy numbers?

Response: In the original reference, rapamycin treatment significantly reduced the number of macrophages (reduced inflammation) and up-regulated LC3 expressions on the injured nerve [2]. To shorten the manuscript, we have deleted the descriptions in such details.

  1. In line 137, what is allodynic response?

Response: “allodynic response” refers to “mechanical allodynia responses.” To shorten the manuscript, we have deleted the descriptions in such details.

  1. The authors mentioned in lines 194-196 that the suppression of apoptosis in dorsal root ganglia can decrease pain!! How you suppress apoptosis, which is a normal physiologic response to injury? How apoptotic cells are cleared? What is the role of autophagy in clearance of apoptotic cells? I recommend adding a paragraph discussing the role of LC3-associated phagocytosis (LAP) in the clearance of apoptotic cells. You can know about it in this article: Eid N, Ito Y. Oxoglaucine alleviates osteoarthritis by activation of autophagy via blockade of Ca2+ influx and TRPV5/calmodulin/CAMK-II pathway. Br J Pharmacol. 2021 Nov 3.

Response: Thanks for your kind reminders. We agreed that the descriptions of “suppression of apoptosis in the dorsal root ganglia can decrease pain” might overexplain the results of previous studies. We have decided to delete those descriptions, and only keep the descriptions of “These studies demonstrated consistent results that apoptotic activities (measured by cleaved caspase-3, cytochrome c, TUNEL assay) on dorsal root ganglia neurons and sat-ellite cell increased from day two to day 28 after nerve injury. They also demonstrated that different agents (including ALCAR, estradiol and recombinant human erythro-poietin) could suppress pain behavior and decrease apoptotic activities in injured dorsal root ganglia at the same time.” In the beginning of part 4, we have added the descriptions and references [3,4] of LC3-associated phagocytosis (LAP) in the clearance of apoptotic cells in the beginning of the section of “Relationships between autophagy and apoptosis in neuropathic pain formation.”

  1. In figure 2, what is the meaning of double head arrows? What are the relations of microautophagy to your story? The diagram is incorrect. The autophagosome fuse with what in left side of the figure?

Response: Thanks for your meticulous reminders. We have modified Figure 2. “double head arrows” are changed to blue or red arrows to indicate activation and inhibition, respectively. “Microautophagy” in the figure 2 is changed to “autolysosome.”. 

Reviewer 2 Report

The Role of Autophagy and Apoptosis in Neuropathic Pain Formation

This manuscript on autophagy and apoptosis in neuro pain relation regulation is a well-themed topic, it requires extensive revision in order to bring a better version of the manuscript, authors must follow up on the recommended points and justify the evaluation.

  1. Improvisation of abstract and introduction is required.
  2. There are syntax, lexical errors, grammatical errors, incomplete sentences, spelling errors especially protein names, spacing errors that must be corrected and edited as per author guideline format.
  3. more references should be included.
  4. A detailed explanation of autophagic and apoptotic roles in neuro pain has been explained which requires a clear representation of inflammatory signaling pathways for neuro pain.
  5. Roles of organelle in inflammatory pain and autophagic and apoptotic induced cell death is a crucial part, in this review, it should be added as per the content.
  6. The subheadings like in 2.2.1-2.2.3 are not at all good, I recommend authors to follow Stiewe et al., 2020 article to understand the pattern of subheadings and context writing format.
  7. Certain formats such as et al., is always Italic, authors must change the format of sentence context in the entire manuscript that is written as author x's or "Kosacka et al.'s studies"
  8. Each of the topics discussed should be a sequential story rather than "X author did these experiments" in the sentences. Therapeutic targets or inhibitors should be included in tables in a separate column
  9. A graphical abstract is essential
  10. Authors should include schematic diagrams of autophagic and apoptotic activity in nerve and brain injury involving regulatory proteins, inhibitors, up and down signals, and agents involved
  11. Fig 1 is not a complete diagram of autophagy, apoptosis in neuro pain, it should include a justified chart of details neuropathic pain lead cell death or recovery and legends to revise well
  12. Fig 2 diagram should indicate autophagic inducers and inhibitors or anti-apoptotic agents and therapeutic targets in crosstalk and pro-inflammatory and inflammatory signals

Author Response

Thanks for your criticisms and constructive suggestions. We believe that those suggestions can significantly improve our manuscript. We made the point-to-point responses to your questions and revised them in the new version of manuscript.

  1. Improvisation of abstract and introduction is required.

Response: Thanks for your suggestions. We have extensively modified the abstract and introduction as reviewer suggested. We have also re-organized the abstract and condensed the introduction. 

  1. There are syntax, lexical errors, grammatical errors, incomplete sentences, spelling errors especially protein names, spacing errors that must be corrected and edited as per author guideline format.

Response: Thanks for your kind reminders. The first edition of our manuscript had been edited by Nature Research Editing Service. However, we will send the final version to MDPI English Editing Services for further English modification before publication.

  1. More references should be included.

Response: Thanks for your suggestions. More references including the relationship between macroautophagy, microautophagy, and chaperone-mediated autophagy (CMA), and the interactions between proinflammatory cytokines, autophagy and apoptosis are added accordingly.

  1. A detailed explanation of autophagic and apoptotic roles in neuro pain has been explained which requires a clear representation of inflammatory signaling pathways for neuro pain.

Response: Thanks for your constructive suggestions. We have added more detailed descriptions about the relationship between autophagy, apoptosis, proinflammatory cytokines and pain in the beginning of the part 4 “Relationships between autophagy and apoptosis in neuropathic pain formation.” 

  1. Roles of organelle in inflammatory pain and autophagic and apoptotic induced cell death is a crucial part, in this review, it should be added as per the content.

Response: We have added the role of lysosome in autophagic processes in the beginning of part 2 “Autophagy in neuropathic pain formation”, and the role of mitochondria in apoptosis in the beginning of part 3 “Apoptosis in neuropathic pain formation” as reviewer suggested.

  1. The subheadings like in 2.2.1-2.2.3 are not at all good, I recommend authors to follow Stiewe et al., 2020 article to understand the pattern of subheadings and context writing format.

Response: Thanks for your kindly suggestions. We have changed the patten of subheadings as reviewer suggested.

  1. Certain formats such as et al.,is always Italic, authors must change the format of sentence context in the entire manuscript that is written as author x's or "Kosacka et al.'s studies"

Response: Thanks for your kind reminders. We have corrected those errors accordingly.

  1. Each of the topics discussed should be a sequential story rather than "X author did these experiments" in the sentences. Therapeutic targets or inhibitors should be included in tables in a separate column

Response: We have re-organized and condensed parts 2 and 3. We have also modified the columns of each table to present the effects therapeutic agents. However, we did not add more columns for individual therapeutic targets or inhibitors to avoid the complexity and too many columns in one table, which may be confusing the readers. 

  1. A graphical abstract is essential

Response: We have modified the Figure 1 and created our graphical abstract. We have added the graphical abstract in the end of abstract accordingly.   

  1. Authors should include schematic diagrams of autophagic and apoptotic activity in nerve and brain injury involving regulatory proteins, inhibitors, up and down signals, and agents involved

Response: We have modified the regarding information in the revised Figure 2. However, we did not add all details into Figure 2 to avoid the complexity in one figure. We have added the alterations of autophagic and apoptotic activities on the injured nerve, dorsal root ganglia, spinal dorsal horn and brain, and the effects of autophagy inducer and suppressor on those regarding changes. 

  1. Fig 1 is not a complete diagram of autophagy, apoptosis in neuro pain, it should include a justified chart of details neuropathic pain lead cell death or recovery and legends to revise well
  2. Fig 2 diagram should indicate autophagic inducers and inhibitors or anti-apoptotic agents and therapeutic targets in crosstalk and pro-inflammatory and inflammatory signals

Response: Thanks for your creative comments and suggestions. Figures 1 and 2 were misplaced. We have replaced the Figures 1 and 2 in the right position. Figure 1 (original Figure 2) referred to the interactions between apoptosis, autophagy, and por-inflammatory cytokines. We have added more detailed descriptions (Blue line indicates activation and red line indicates inhibition) in Figure 1 (original Figure 2). We have also added more details including the effects of autophagy inducer and suppressor in Figure 2. However, to avoid the complexity in one single figure, we decide not to add all autophagic inducers/inhibitors, and anti-apoptotic agents in the Figure 1 (original Figure 2). We apologize for those regards. 

Reviewer 3 Report

The manuscript performed by Ming-Feng Liao and colleagues aims to summarize the implication of autophagy and apoptosis in neuropathic pain formation.

The manuscript needs to be improved and address some issues before publication:

Abstract:

  • Line 14: “the somatosensory sensory system” sensory is repeated twice.
  • Line 19: “apoptotic activities have adverse effects on neuropathic pain formation”, this sentence, to my understanding, means that apoptosis has negative effects on pain formation, thus, apoptosis if beneficial. However, in lines 21-22, the authors state “Agents that can enhance autophagic activities but suppress apoptotic activities on the injured nerve and dorsal root ganglia have the potential to treat neuropathic pain.” Which is contradictory.
  • Lines 22-23: “different studies have shown inconsistent results in apoptotic and autophagic activities in the spinal cord after nerve damage.” Right after the statement that apoptosis must be suppressed, the authors say that there is some controversy about it. Authors should reorganize their abstract: firstly, there is a controversy regarding apoptosis involvement in nerve pain formation, then say that they summarized the different findings and concluded that apoptosis is negative, and at the end, suggest the pro-autophagic & anti-apoptotic treatment.

Main text:

  • Line 92: “However, some authors stated that increased LC3-II levels indicated delayed autophagosome clearance and downregulated autophagic processes.” Please add reference.
  • “rapamycin (autophagy inducer)” is repeated in lines 126 and 130 and 132 and 142 and then in page 6…. Please, mention it only ONCE.
  • “3-MA (autophagy inhibitor)” is repeated in lines 131 and 143, and then in page 6, then page 14….
  • Lines 8-10 page 6: “Other studies utilized indirect methods; researchers monitored the effects of different proinflammatory mediators on pain behavior and autophagic protein expression in the spinal dorsal horn” Please add reference.
  • Titles 2.2.1, 2.2.2 and 2.2.3 are so confusing! It looks like where is 2.2.4 to have all the possibilities of word combinations? Please, simplify this, I would rather use “cases where autophagy acts as a pain suppressor” and “cases where autophagy acts as a pain enhancer”.
  • Regarding autophagy; the authors do not distinguish among the three types of autophagy in any of their examples. Could it be that one type of autophagy is inhibited while another type is promoted? Is there any preference for one type of autophagy or another in nerve pain generation? The authors should address this topic in the text.
  • Lines 19: “Autophagy and apoptosis have crosstalk under stress conditions, basically through 45 mutual inhibition [6].” This is not true as it has been demonstrated that autophagy can also promote cell apoptotic death.
  • In general, parts 2 and 3 must be shortened. The most Important is part 4 which arrives at page 19.
  • Figure 1: where are the proinflammatory cytokines in the figure??
  • Authors should add a part of the relation ship between proinflammatory cytokines with both autophagy and apoptisis.

English:

  • The text needs to be carefully revised to correct all the spelling and grammar mistakes.

Author Response

Thanks for your detailed and constructive suggestions. We believe that those criticisms and suggestions can significantly improve our manuscript. We made the point-to-point responses to your questions and revised them in the new version of manuscript.

Abstract:

  • Line 14: “the somatosensory sensory system” sensory is repeated twice.

Response: we have deleted this redundant word.

  • Line 19: “apoptotic activities have adverse effects on neuropathic pain formation”, this sentence, to my understanding, means that apoptosis has negative effects on pain formation, thus, apoptosis if beneficial. However, in lines 21-22, the authors state “Agents that can enhance autophagic activities but suppress apoptotic activities on the injured nerve and dorsal root ganglia have the potential to treat neuropathic pain.” Which is contradictory.
  • Lines 22-23: “different studies have shown inconsistent results in apoptotic and autophagic activities in the spinal cord after nerve damage.” Right after the statement that apoptosis must be suppressed, the authors say that there is some controversy about it. Authors should reorganize their abstract: firstly, there is a controversy regarding apoptosis involvement in nerve pain formation, then say that they summarized the different findings and concluded that apoptosis is negative, and at the end, suggest the pro-autophagic & anti-apoptotic treatment.

Response to lines 19 and 22-23: Thanks for your kind reminders and suggestions about the abstract. We have deleted the descriptions of “apoptotic activities have adverse effects on neuropathic pain formation”. We have re-organized the sentences that can avoid the misleading meaning in the abstract.

Main text:

  • Line 92: “However, some authors stated that increased LC3-II levels indicated delayed autophagosome clearance and downregulated autophagic processes.” Please add reference.

Response: Thanks for your kind reminders. We have added the reference [1]. The authors suggested that lately increased LC3-II expressions indicating autophagic flux was blocked and concluded that autophagic activities decreased.

  • “Rapamycin (autophagy inducer)” is repeated in lines 126 and 130 and 132 and 142 and then in page 6…. Please, mention it only ONCE.

Response: Thanks for your kind reminders. We have deleted the redundant words accordingly.

  • “3-MA (autophagy inhibitor)” is repeated in lines 131 and 143, and then in page 6, then page 14….

Response: Thanks for your reminders. We have deleted the redundant words accordingly.

Lines 8-10 page 6: “Other studies utilized indirect methods; researchers monitored the effects of different proinflammatory mediators on pain behavior and autophagic protein expression in the spinal dorsal horn” Please add reference.

Response: Thanks for your reminders. We have added references [1-6] following the sentences.

  • Titles 2.2.1, 2.2.2 and 2.2.3 are so confusing! It looks like where is 2.2.4 to have all the possibilities of word combinations. Please, simplify this, I would rather use “cases where autophagy acts as a pain suppressor” and “cases where autophagy acts as a pain enhancer”.

Response: Thanks for your suggestions. We have changed and simplified the subtitles as “2.2.1 Increased autophagic activities in the spinal cord neurons, decreased autophagic activities in the spinal cord microglia and astrocytes after nerve injury, and autophagy acts as a pain suppressor”, and “2.2.2 Increased autophagic activities in the spinal cord neurons and microglia after nerve injury, and autophagy acts as a pain enhancer.”

  • Regarding autophagy, the authors do not distinguish among the three types of autophagy in any of their examples. Could it be that one type of autophagy is inhibited while another type is promoted? Is there any preference for one type of autophagy or another in nerve pain generation? The authors should address this topic in the text.

Response: Thanks for your suggestions. We have added more detailed descriptions of the interaction of three types of autophagy in the beginning of part 2 “autophagy in neuropathic pain formation”: “There are three types of autophagy: macroautophagy, microautophagy, and chaperone-mediated autophagy (CMA)[7]”, “Micro-autophagy needs to be studied by electron microscopy and is still obscure in mammalian cells [8]. On the other hand, CMA is observed only in mammalian cells, and has protein markers including heat shock-cognate chaperone of 70 kDa (HSC70), and lysosome-associated membrane protein type-2A (LAMP2A) [8,9]. Both macro-autophagy and CMA play the essential roles of central neurological degeneration diseases [10]. CMA pathway compensates when the macro-autophagy processes is inhibited; in contrast, CMA block induces compensative upregulation of macro-autophagy [10]. Many studies focused on the macroautophagy changes on the neurons of the animals after the nerve damages with rapamycin or 3 -MA treatment. Most studies proposed that when macroautophagic activities increase, the expression levels of LC3-II, Bclin-1, and Atg5-Atg12 increase, but p62 levels (delivering ubiquitinated cargoes to autophagosomes) decrease[2-6,11-24].

Macroautophagy.

  • Lines 19: “Autophagy and apoptosis have crosstalk under stress conditions, basically through 45 mutual inhibition [6].” This is not true as it has been demonstrated that autophagy can also promote cell apoptotic death.

Response: Thanks for your kind reminder. We have changed the descriptions to “Autophagy and apoptosis have crosstalk through various proteins including Atg5–Atg12, Beclin-1/Bcl-2, and caspase-3 under stress conditions.” 

  • In general, parts 2 and 3 must be shortened. The most Important is part 4 which arrives at page 19.

Response: Thanks for your constructive suggestions. We have modified and shortened the parts 2 and 3, and have added more discussions including the relationship between proinflammatory cytokines with both autophagy and apoptosis in part 4.

  • Figure 1: where are the proinflammatory cytokines in the figure??

Response: Thanks for your kind reminder. Figure 1 and Figure 2 were misplaced. We have corrected the Figures 1 and 2 in the right position. Figure 1 referred to the interactions between apoptosis, autophagy, and por-inflammatory cytokines. We have added more detailed descriptions in Figure 1 (Blue line indicates activation, and red line indicates inhibition).

  • Authors should add a part of the relationship between proinflammatory cytokines with both autophagy and apoptosis.

Response: Thanks for your positive and constructive suggestions. We have added the relationship between proinflammatory cytokines, autophagy and apoptosis in the part 4 “Relationships between autophagy and apoptosis in neuropathic pain formation” in the Discussion.

“Autophagy and apoptosis have crosstalk through different proteins, including Atg5–Atg12, Beclin-1/Bcl-2, and caspase-3 under stress conditions, basically through mutual inhibition. Besides, a non-classical autophagy process called LC3-associated phagocytosis (LAP) can help remove apoptotic cells by macrophages and inhibit proinflammatory processes. In addition, autophagy and apoptosis are both regulated by proinflammatory cytokines. Autophagy has the strong interactions with proinflammatory and anti-inflammatory cytokines, mainly in immune cells including lymphocytes and macrophages. Proinflammatory cytokines, including IFN-γ, TNF-α, IL-1β, and IL-6 have been shown to have the abilities to activate autophagic processes, while the anti-inflammatory cytokines including IL-4 and IL-10 could inhibit autophagic activities. Comparatively, autophagy could suppress activities of proinflammatory cytokines, mainly IL-1 family, but also IL-6, TNF-α, and IL-17. Proinflammatory cytokines could directly activate apoptosis in nonneuronal cells, including β-cells and epithelial cells. The contributions of proinflammatory cytokines including IL-1 and TNF-α to apoptosis in neurodegenerative diseases such as Alzheimer’s disease and amyotrophic lateral sclerosis are also well established. In addition to autophagy and apoptosis, proinflammatory cytokines play important roles in neuropathic pain formation, and thus, inhibiting proinflammatory cytokines may help to attenuate neuropathic pain. Therefore, proinflammatory cytokines could be the key regulators between autophagy and apoptosis in neuropathic pain formation[25-30].

English:

  • The text needs to be carefully revised to correct all the spelling and grammar mistakes.

Response: Thanks for your kind reminders. The first edition of our manuscript had been edited by Nature Research Editing Service. However, we will send the final version to MDPI English Editing Services for further English modification before publication.

Round 2

Reviewer 2 Report

2nd review recommendation

In order to bring a better version of the manuscript, authors must follow up on the recommended points and justify the evaluation

  1. Recommended of 1st review comment no 10-11. Fig 1 of the old manuscript now fig 2 in the revised image indicates autophagic inducer and inhibitor or anti-apoptotic agent. Now the image and chart in the image representing did not fit the image given. Figure image of the brain, spinal cord, peripheral nerve, dorsal root ganglia should indicate the mark location of autophagy and apoptosis in neuro pain or injury situations. Image graphic should indicate neurons, Schwann cells, axon, satellite cells, microglia cells, astrocytes location, and after-effects of autophagy and apoptosis. Example –normal to neuro injury or pain mark, transition from neuro pain site of apoptosis to autophagy to pain reduction location. A detailed figure legend is not necessary, legend should be precise enough of explaining the image representation. Figure image, should be explained in results or discussion and not in legends. Authors must check for spelling-grammatical-manuscript font style errors in the image and alignment.
  1. Marking figure 2, in conclusion, is irrelevant. Line 104-116 does not justify anything as per the context. Need a serious revision to write the conclusion.
  2. Figure 2 of the revised manuscript legend should be crosstalk between autophagy and apoptosis during neuropathic pain. The image should include autophagy, apoptosis and inflammatory proteins and type of neuronal inflammation and the animal models reported.
  3. Revised abstract requires more precision. Line 30-32, must be corrected as -this review may help to understand the treatment strategy of neuropathic pain through modulating proinflammatory cytokines, apoptotic and autophagic …
  4. 1st review comment no 8, mentioned to authors for the story of content to write instead of writing Author X et al’s studies demonstrated that in the explanation, which is not followed up well.
  5. Apoptotic activity changes in the injured nerve and dorsal root ganglia after nerve injury or Apoptotic activity changes in the spinal cord after nerve injury – still not convenient with the context. It should be like a story format and not reporting of referencing article or author’s work and similar for the autophagy details given here.
  6. Name of mice and rat animal model should be included in text and table if available
  7. Line 10-11 – while suggesting or justifying the sentence should be thus - this hypothesized that autophagic activity can affect neuropathic pain indirectly
  8. Introduction requires more explanation and an organized way of pointing out the content.
  9. There are syntax, lexical errors, grammatical errors, incomplete sentences, severe spelling and spacing errors to revise by the authors in the manuscript, must check all protein names.
  10. In many sentences it is mentioned by authors as day one or two and so to day 28, Authors must follow one format like day 1 to 28 in numbers or in alphabets
  11. Insufficient references in the manuscript

Author Response

Thanks for your constructive comments and suggestions. We believe that those comments can improve the quality of our manuscript. Here, we have made the point-to-point responses to your suggestions and revised them in our new version of manuscript.

  1. Recommended of 1streview comment no 10-11. Fig 1 of the old manuscript now fig 2 in the revised image indicates autophagic inducer and inhibitor or anti-apoptotic agent. Now the image and chart in the image representing did not fit the image given. Figure image of the brain, spinal cord, peripheral nerve, dorsal root ganglia should indicate the mark location of autophagy and apoptosis in neuro pain or injury situations. Image graphic should indicate neurons, Schwann cells, axon, satellite cells, microglia cells, astrocyte’s location, and after-effects of autophagy and apoptosis. Example –normal to neuro injury or pain mark, transition from neuro pain site of apoptosis to autophagy to pain reduction location. A detailed figure legend is not necessary, legend should be precise enough of explaining the image representation. Figure image, should be explained in results or discussion and not in legends. Authors must check for spelling-grammatical-manuscript font style errors in the image and alignment.
  2. Marking figure 2, in conclusion, is irrelevant. Line 104-116 does not justify anything as per the context. Need a serious revision to write the conclusion.
  3. Figure 2 of the revised manuscript legend should be crosstalk between autophagy and apoptosis during neuropathic pain.

The image should include autophagy, apoptosis and inflammatory proteins, and type of neuronal inflammation and the animal models reported.

Response: Thanks for your comments and suggestions about Figure 2. We have made extensive modifications of Figure 2 and its related legend according to the suggestions.

  1. We have marked each table to correlative anatomical-locations of the injured nerve, dorsal root ganglia, spinal cord, and brain.
  2. Axons, Schwann cells, dorsal root ganglia neurons, satellite cells, microglia cells, astrocytes are listed on individual column and described their changes after various nerve injury with different animal models in details.
  3. The legends of the figures and conclusion are revised in the new version of manuscript. The legends of the figures were more condensed as reviewer suggested. The conclusion part was rewritten to fit the content of Figure 2. Spelling and grammar errors were rechecked and revised accordingly.
  4. The changes of autophagic, apoptotic, proinflammatory agent activities and their relationships were described in details in each column.
  5. Different animal models used in those studies were also listed in the tables.
  1. Revised abstract requires more precision. Line 30-32, must be corrected as -this review may help to understand the treatment strategy of neuropathic pain through modulating proinflammatory cytokines, apoptotic and autophagic.

Response: Thanks for your suggestions. We have revised the descriptions on lines 30-32 according to reviewer’s suggestions.

  1. 1streview comment no 8, mentioned to authors for the story of content to write instead of writing Author X et al’s studies demonstrated that in the explanation, which is not followed up well.
  2. Apoptotic activity changes in the injured nerve and dorsal root ganglia after nerve injury or Apoptotic activity changes in the spinal cord after nerve injury – still not convenient with the context. It should be like a story format and not reporting of referencing article or author’s work and similar for the autophagy details given here.

Response: Thanks for your kind reminds. We have corrected and avoided the descriptions of “authors X et al’s studies showed:” in the manuscript including the section of “2.3. Autophagic activity changes in the brain after nerve injury.” “3.2. Apoptotic activity changes in the spinal cord after nerve injury.” and “4. Relationships between autophagy and apoptosis in neuropathic pain formation.” We hope the revised version of manuscript is more like a story format and is more readable for the readers.

  1. Name of mice and rat animal model should be included in text and table if available

Response: Thanks for your constructive suggestions. We have added the animal models used in the texts and tables in the revised manuscript that will be more precise descriptions in a review article.

  1. Line 10-11 – while suggesting or justifying the sentence should be thus - this hypothesized that autophagic activity can affect neuropathic pain indirectly

Response: Thanks for your suggestions. We have revised the descriptions on lines 10-11 according to reviewer’s suggestions.

  1. Introduction requires more explanation and an organized way of pointing out the content.

Response: Thanks for your constructive suggestions. We have added more descriptions in the introduction, mainly the discussion of the relationships between autophagy, apoptosis, and proinflammatory cytokines that were not described before. We hope the revised version of manuscript will be more organized and readable.   

  1. There are syntax, lexical errors, grammatical errors, incomplete sentences, severe spelling and spacing errors to revise by the authors in the manuscript, must check all protein names.

Response: Thanks for your meticulous reminds. Spelling, grammatical, and spacing errors were checked carefully once more. We will send the final version to MDPI English Editing Services before publication if necessary.

  1. In many sentences it is mentioned by authors as day one or two and so to day 28, Authors must follow one format like day 1 to 28 in numbers or in alphabets

Response: Thanks for your kind reminds. We have changed the “day one” to “day 1” in the manuscript that will be in a consistent format.

  1. Insufficient references in the manuscript

Response: Thanks for your kind reminds. We have added more references to each description in the manuscript that will strengthen the evidence.

Reviewer 3 Report

The manuscript looks way better now.

Author Response

Thanks for your comments and encouragement. We appreciate your constructive suggestions that make the revised manuscript much better.

Round 3

Reviewer 2 Report

Some of the recommended revision has been done well in this manuscript still major revision is needed.

3rd round review comment has been given -

Authors must read well each concerned comments to understand and should follow the manuscript guidelines

Fig 2 image has represented well but requires clean construction.

  1. Labels should not overlap with each other or with figures in image
  2. Font check with text format
  3. Indicated tables can be well constructed rather than simple blue shades table style in the background
  4. Indicative arrows must be used for marking the location
  5. Fig 2 legend title recommended - Schematic diagram showing the underlying mechanism of autophagic and apoptotic activities in the nerve injury.

Already recommended, Fig 2 legend- is exaggerated from line 617-717 and any discussion with previously reported study should be done in the text and not in conclusion.

Conclusions and Perspectives of previous review

Most studies have shown consistent findings that autophagic and apoptotic activities in injured nerves and dorsal root ganglia increase after nerve damage. Apoptotic activities of the injured nerve and dorsal root ganglia elevated after nerve injury. However, there is less convincing evidence demonstrating that the pain behavior could be suppressed by the agents that directly inhibit apoptosis. On the other hand, more evidence has shown that the upregulated autophagic activities on the injured nerve and dorsal root ganglia after nerve damage could possibly help myelin clearance and promote nerve regeneration, which can attenuate pain behavior. Agents that can suppress promote autophagic activities, or apoptotic activities or promote autophagic activities on the injured nerve and dorsal root ganglia can potentially treat neuropathic pain. However, the autophagic and apoptotic changes in the spinal cord after peripheral nerve injury have shown inconsistent results in different studies (Figure 2). Complex central sensitization mechanisms in the spinal cord, different animal models, different study designs and times, and the use of various  treatment agents can cause these inconsistent findings. Nevertheless, alterations in autophagic and apoptotic activities over time in the spinal cord play essential roles in neuropathic pain formation after nerve damage. The crosstalk or interaction of autophagy and apoptosis in the spinal cord neuropathic pain formation occurs mainly through modulation of pro-inflammatory cytokines. Further detailed studies of autophagy, apoptosis, pro-inflammatory cytokines and their interactions in the neuropathic pain formation spinal cord after nerve injury are warranted to develop a new strategy to treat neuropathic pain.

The red-inked lines indicated to revise, but instead, it is completely changed, and why references are cited in the conclusion? Authors should not be confused at all in the pattern of conclusion? I recommend revising these marked lines of conclusion as per comment and no reference in conclusion and adding the line – of recent revision - the alterations of autophagic/apoptotic activities and their interactions with pro-inflammatory agents in the injured nerve, dorsal root ganglia, spinal cord, and brain were summarized in Figure 2.

in abstract syntax, lexical error, grammatical errors to revise well. Revision line recommended - This review may help to understand the treatment strategy of neuropathic pain during nerve injury through modulating apoptotic/autophagic activities and pro-inflammatory cytokines interactions in the nervous system.

Graphical abstract, image could be modified with colors and design and title recommended– Interaction chart of autophagy, apoptosis, pro-inflammatory cytokines during neuropathic pain in the nerve injury.

Table 1, Table 2 and Table 4 title should be revised as different models of rat and mice. Table 3 should be revised as different rat models. **Tables texts should be rechecked for plagiarism

Introduction line 77-78 should be changed to authors own words. Neuropathic pain is defined as "pain caused by a lesion or disease of the somatosensory nervous system" by the International Association for the Study of Pain (IASP) [1]. (plagiarism)

Figure 1. legend title recommended -Crosstalk between autophagy, apoptosis, and proinflammatory cytokines during in neuropathic pain formation

**References format is not similar with the text

***There are still severe syntax, lexical errors, grammatical errors, spacing and alignment (table texts are not unique) errors. Major English correction recommended

Author Response

Response to reviewer 2:

Thanks for your detailed reviews and suggestions. We believe that those suggestions can improve the quality of our manuscript. Here, we have revised them in our final version of the manuscript. We have also sent our manuscript to MDPI English editing for grammatical and spelling errors corrections. Down below are our point-to-point responses to your reminders and suggestions.

Fig 2 image has represented well but requires clean construction.

  1. Labels should not overlap with each other or with figures in image
  2. Font check with text format
  3. Indicated tables can be well constructed rather than simple blue shades table style in the background
  4. Indicative arrows must be used for marking the location
  5. Fig 2 legend title recommended - Schematic diagram showing the underlying mechanism of autophagic and apoptotic activities in the nerve injury.

Already recommended, Fig 2 legend- is exaggerated from line 617-717 and any discussion with previously reported study should be done in the text and not in conclusion.

Response: Thanks for your detailed suggestions about Figure 2. We have modified Figure 2 and its related legend according to the recommendations.

  1. We have modified the tables and labels to avoid labels overlapping with figures in the image.
  2. The contents of the tables have been reorganized and the edges of each table are enhanced with black lines to make them well constructed and more readable.
  3. We have used indicative arrows for marking the anatomical locations of Figure 2.
  4. We have modified the title of Figure 2 in the following sentence: "Schematic diagram showing the underlying mechanisms of autophagic and apoptotic activities in the nerve injury." according to the reviewer’s suggestions.
  5. Text format and their font in the tables are carefully checked and revised accordingly.
  6. The legend of Figure 2 is simplified, and the content of the conclusion is simultaneously revised.

Conclusions and Perspectives of previous review

The red-inked lines indicated to revise, but instead, it is completely changed, and why references are cited in the conclusion? Authors should not be confused at all in the pattern of conclusion? I recommend revising these marked lines of conclusion as per comment and no reference in conclusion and adding the line – of recent revision - the alterations of autophagic/apoptotic activities and their interactions with proinflammatory agents in the injured nerve, dorsal root ganglia, spinal cord, and brain were summarized in Figure 2.

Response: Thanks for your comments and suggestions. We have revised the content and removed the citations in the conclusion. We have also added the description of "the alterations of autophagic/apoptotic activities and their interactions with proinflammatory agents in the injured nerve, dorsal root ganglia, spinal cord, and brain are summarized in Figure 2" in conclusion.

in abstract syntax, lexical error, grammatical errors to revise well. Revision line recommended - This review may help to understand the treatment strategy of neuropathic pain during nerve injury through modulating apoptotic/autophagic activities and proinflammatory cytokines interactions in the nervous system.

Response: Thanks for your suggestions. We have revised syntax, lexical error, grammatical errors according to your suggestions. And all revisions are marked with lines. The abstract is summarized as the following sentence: “This review may help in further understanding the treatment strategy of neuropathic pain during nerve injury by modulating apoptotic/autophagic activities and proinflammatory cytokines in the nervous system.”  

Graphical abstract, image could be modified with colors and design and title recommended– Interaction chart of autophagy, apoptosis, proinflammatory cytokines during neuropathic pain in the nerve injury.

Response: Thanks for your suggestions. We have modified the colors and design in the image. We have also changed the title of the graphical abstract to “Interaction chart of autophagy, apoptosis, proinflammatory cytokines during neuropathic pain in the nerve injury” according to your recommendations.

Table 1, Table 2, and Table 4 title should be revised as different models of rat and mice. Table 3 should be revised as different rat models. **Tables texts should be rechecked for plagiarism

Response: Thanks for your suggestions. We have changed the titles of tables 1, 2, and 4 to different models of rats and mice, and table 3 to different rat models as the reviewer’s suggestions. We have also sent our manuscript to the MDPI English editing service to help rewrite and avoid plagiarism.

Introduction line 77-78 should be changed to authors own words. Neuropathic pain is defined as "pain caused by a lesion or disease of the somatosensory nervous system" by the International Association for the Study of Pain (IASP) [1]. (plagiarism)

Response: Thanks for your reminders and suggestions. To avoid the possibility of plagiarism, we have changed the description to " Neuropathic pain is induced by damage to the somatosensory nervous system."

Figure 1. legend title recommended -Crosstalk between autophagy, apoptosis, and proinflammatory cytokines during in neuropathic pain formation

Response: Thanks for your suggestions. We have revised the legend title of Figure 1 to “Crosstalk between autophagy, apoptosis, and proinflammatory cytokines during neuropathic pain formation” as your suggestions.

**References format is not similar with the text

Response: Thanks for your friendly reminders. We have corrected the format of the references and made it consistent with the text format.

***There are still severe syntax, lexical errors, grammatical errors, spacing and alignment (table texts are not unique) errors. Major English correction recommended

Response: Thanks for your suggestions. We have sent the manuscript to the MDPI English editing service to help correct syntax, lexical errors, grammatical errors, spacing, and alignment errors.
